# Reprogramming Megakaryocytes for Controlled Release of Platelet-like Particles Carrying a Single-Chain Thromboxane A_2_ Receptor-G-Protein Complex with Therapeutic Potential

**DOI:** 10.3390/cells12242775

**Published:** 2023-12-06

**Authors:** Renzhong Lu, Yan Li, Anna Xu, Bridgette King, Ke-He Ruan

**Affiliations:** The Center for Experimental Therapeutics and Pharmacoinformatics, Department of Pharmacological and Pharmaceutical Sciences, College of Pharmacy, University of Houston, Houston, TX 77204, USA; rlu12@cougarnet.uh.edu (R.L.); yanli93776@gmail.com (Y.L.); axu2@wellesley.edu (A.X.); king616@marshall.edu (B.K.)

**Keywords:** platelet-linked particles, megakaryocyte, G-protein-coupled receptor

## Abstract

In this study, we reported that novel single-chain fusion proteins linking thromboxane A_2_ (TXA_2_) receptor (TP) to a selected G-protein α-subunit q (SC-TP-Gαq) or to α-subunit s (SC-TP-Gαs) could be stably expressed in megakaryocytes (MKs). We tested the MK-released platelet-linked particles (PLPs) to be used as a vehicle to deliver the overexpressed SC-TP-Gαq or the SC-TP-Gαs to regulate human platelet function. To understand how the single-chain TP-Gα fusion proteins could regulate opposite platelet activities by an identical ligand TXA_2_, we tested their dual functions—binding to ligands and directly linking to different signaling pathways within a single polypeptide chain—using a 3D structural model. The immature MKs were cultured and transfected with cDNAs constructed from structural models of the individual SC-TP-Gαq and SC-TP-Gαs, respectively. After transient expression was identified, the immature MKs stably expressing SC-TP-Gαq or SC-TP-Gαs (stable cell lines) were selected. The stable cell lines were induced into mature MKs which released PLPs. Western blot analysis confirmed that the released PLPs were carrying the recombinant SC-TP-Gαq or SC-TP-Gαs. Flow cytometry analysis showed that the PLPs carrying SC-TP-Gαq were able to perform the activity by promoting platelet aggregation. In contrast, PLPs carrying SC-TP-Gαs reversed Gq to Gs signaling to inhibit platelet aggregation. This is the first time demonstrating that SC-TP-Gαq and SC-TP-Gαs were successfully overexpressed in MK cells and released as PLPs with proper folding and programmed biological activities. This bio-engineering led to the formation of two sets of biologically active PLP forms mediating calcium and cAMP signaling, respectively. As a result, these PLPs are able to bind to identical endogenous TXA_2_ with opposite activities, inhibiting and promoting platelet aggregation as reprogrammed for therapeutic process. Results also demonstrated that the nucleus-free PLPs could be used to deliver recombinant membrane-bound GPCRs to regulate cellular activity in general.

## 1. Introduction

Recombinant proteins have become increasingly popular for use in therapeutics. Recombinant protein drugs make up 23% of the total drugs approved by the FDA in 2020 [1]. The majority of recombinant proteins suitable for drug use are limited to soluble proteins, such as antibodies, insulin, and growth factors. Membrane-bound proteins, such as G-protein-coupled receptors (GPCRs) are not compatible for use as injectable protein drugs due to their insoluble nature and tendency to aggregate. Isolated GPCRs also lack the necessary ability of anchoring to cell membranes coupled with signaling molecules in a live cell environment. Cellular delivery of membrane receptors could provide an alternative method to advance recombinant protein therapy in the near future.

Currently, GPCRs are not able to be engineered into an injectable protein drug. However, GPCRs are an important member of a family of membrane proteins involved in performing and maintaining normal life for almost all cells and organs [2,3,4,5,6,7]. The critical role of GPCRs is further highlighted by the fact that deficiencies or dysfunctions in these receptors are implicated in a wide range of human diseases [8,9]. Such diseases can include neurological disorders, cardiovascular conditions, metabolic syndromes, and various cancers [10,11,12,13]. Consequently, GPCRs have been the focus of extensive research efforts in drug discovery and development. Numerous drugs have been designed and developed by specifically targeting functional GPCRs [14]. These drugs can either activate or inhibit GPCR signaling pathways, depending on the therapeutic goal. For example, beta-adrenergic receptors, a type of GPCR, are targeted by beta-blockers to treat conditions like hypertension and cardiac arrhythmias [10]. Similarly, serotonin receptors are targeted by selective serotonin reuptake inhibitors (SSRIs) to manage mood disorders like depression and anxiety [15].

Platelet-like particles (PLPs) prepared from immature megakaryocytes (MKs) using growth factor induction in vitro have been reported [16]. PLPs are nucleus-free cell-like particles suitable for delivery from local injection to IV infusion. Expression of the recombinant proteins anchored on PLP membranes could turn PLPs into a novel delivery vehicle of membrane-bound proteins for therapeutics. Therefore, we hypothesize that the PLPs released from the cDNA-transfected MKs, which express the GPCRs, could carry the biological function of recombinant membrane proteins. To test this hypothesis, we focused on identifying PLPs carrying recombinant GPCR functions.

Recently, we developed a “biolink” approach, which engineered several membrane-bound fusion proteins from enzymes to GPCRs [17,18,19,20]. The fusion proteins, single-chain (SC) GPCR-Gα complexes: thromboxane A_2_ (TXA_2_) receptor (TP) covalently linked to G-protein αq (SC-TP-Gαq) and αs (SC-TP-Gαs) with the ability to bind TXA_2_ and perform signaling with a single molecule, were the first successfully to be created and characterized by our group [20]. HEK293 cells regulated cellular signaling in two ways—either by Gαq-mediated calcium signaling or by Gαs-mediated cAMP signaling upon binding to the identical TXA_2_ agonists [20]. In this study, we used SC-TP-Gαq and SC-TP-Gαs as examples to test the hypothesis, in which the PLP system could be used as a novel nucleus-free vehicle to deliver the membrane-bound GPCR-Gα complex with therapeutic potentials. The study demonstrated that the PLPs produced from the Meg-01 cells transfected with the cDNA of SC-TP-Gαq or SC-TP-Gαs were able to carry the expressed SC-TP-Gαq or SC-TP-Gαs proteins by controlling Gq-mediated human platelet aggregation and Gs-mediated anti-human platelet aggregation.

## 2. Materials and Methods

### 2.1. Materials

Meg-01 cell lines were purchased from ATCC (Manassas, VA, USA). Cell culture medium, supplements, and antibiotics were purchased from Life Technologies (Grand Island, NY, USA). Thrombopoietin (TPO) was purchased from R&D systems (Cat#288-TPN-010/CF, Minneapolis, MN, USA). Gαq and Gαs antibodies were purchased from Cayman (Ann Arbor, MI, USA), and CD41a (Cat#11-0411-82) and CD42b (Cat#12-0429-42) antibodies were purchased from Invitrogen (Carlsbad, CA, USA).

### 2.2. Cell Culture

Meg-01 cells were cultured in RPMI 1640 medium (Cat#30-2001, ATCC) and supplemented with 10% fetal bovine serum (FBS) and 1% Antibiotic—Antimycotic at 37 °C in a humidified 5% CO_2_ incubator. Because a suspension cell line was used, the subculture was centrifuged at a speed of 200× *g* for 5 min. The cell pellet was then resuspended in fresh culture medium.

### 2.3. Designing SC-TP-Gαq and SC-TP-Gαs

The functional SC-TP-Gαq and SC-TP-Gαs were created by linking the C-terminus of the human TP (alpha isoform) to the N-terminus of the Gαq or Gαs in a computational simulation using the crystal structures of human TP, Gαq, and Gαs [21]. 

### 2.4. cDNA Synthesis and Plasmid Preparation

The cDNA, in which the C-terminus of the TP was linked to the N-terminus of the Gαq subunit (SC-TP-Gαq) or the Gαs subunit (SC-TP-Gαs), was created using the PCR approach. Both full sizes of the cDNAs were further sub-cloned into a pcDNA3.1 vector using Ecor1 and Xho1 restriction sites, respectively [20].

### 2.5. Expression of the Recombinant SC-TP-Gαq and SC-TP-Gαs in MK Cells

The method to establish stable cell lines expressing the SC-TP-Gαq and SC-TP-Gαs in MK cells is similar to that previously described in HEK293 cells [21]. Briefly, MK cells cultured in suspension medium were transfected with the cDNA vectors by using an electroporation approach detailed by the electroporator manufacturer (ECM 830 square wave electroporation system, BTX, NY, USA). After 48 h of transfection, G-418 (Cat#4727878001) (400 µg/mL) was added to the medium to screen and select for MK cells expressing the recombinant proteins. The selection lasted 3–8 weeks.

### 2.6. Western Blot

The Western blot procedure followed the protocol provided by the Abcam (Cambridge, MA, USA) website. The cells were washed with cold PBS, scratched, and harvested into a 1.5 mL Eppendorf tube. They were collected by centrifugation, and a 200 µL lysis buffer (1× RIPA buffer, 1 mM PMSF, 1× Protease inhibitor) was added and mixed with the cell pellet. The cell lysates were incubated on ice for 30 min while vortexing on and off. After vortexing, they were centrifuged at 16,000 rpm for 20 min at 4 °C. The supernatant was decanted into a new tube. A quantity of 10 µL of lysate was used to perform a protein assay (Pierce 660 nm Protein Assay, Thermo Scientific, Waltham, MA, USA). A quantity of 30 µg of protein was subjected to electrophoresis with lab-made gradient SDS-PAGE gel (4% stacking gel and 10% separation gel). The samples on the gel were transferred onto a PVDF membrane (Bio-Rad, Hercules, CA, USA). COX-1, and Gαs and Gαq antibodies were diluted at 1:300 in 1% bio-grade milk. β-actin and secondary antibodies from ThermoFisher Scientific were diluted at 1:2500 in 1% bio-grade non-fat milk.

### 2.7. Flow Cytometry Analysis

A sensitive and reliable flow cytometry-based platelet aggregation assay was further modified and used to determine the PLP activity [22]. Either the platelet-rich plasma (PRP) or PLP was incubated with the FITC-conjugated anti-CD41a antibody or APC-conjugated anti-CD42b antibody for 10 min at a dilution ratio of 1:10. The stained samples were mixed gently, and arachidonic acid (AA) was added to a final concentration of 3 µM to produce TXA_2_. After 10 min, the sample was injected into the flow cytometry machine for analysis of the TXA_2_-induced platelet aggregation through SC-TP-Gαq signaling and anti-platelet aggregation through SC-TP-Gαs signaling.

## 3. Results

### 3.1. Bio-Engineering of SC-TP-Gαq and SC-TP-Gαs for Controlling of Platelet Functions

TXA_2_, produced by activated platelets, is a potent endogenous factor involved in platelet aggregation and vasoconstriction. TXA_2_ engages in high-affinity binding with the extracellular domain of its GPCR TP, leading to the formation of a non-covalent complex with the intracellular Gαq subunit. This signaling pathway is directly involved in blood clotting, thrombosis, and stroke process. Under physiological conditions, the actions of TXA_2_ are counteracted by PGI_2_. PGI_2_ binds to another GPCR IP and initiates intracellular signaling through Gαs, triggering cAMP signaling. As a result, this leads to anti-platelet aggregation and vasodilation. The two ligands (TXA_2_ and PGI_2_) and two receptors (TP and IP) are directly involved in regulating platelet and vascular functions. Recently, we developed a method using a single ligand, TXA_2_, to control both Ca^++^ and cAMP signaling through the newly engineered SC-TP-Gαq and SC-TP-Gαs complexes, expressed in HEK293 cells [20]. However, the previous study using cancer cell lines expressing SC-TP-Gαq and SC-TP-Gαs do not have any therapeutic potential. In this study, we further describe the details of the SC-TP-Gα design and use MK-PLP cells with therapeutic potential to carry either the overexpressed SC-TP-Gαq or SC-TP-Gαs to regulate platelet functions (Figure 1).

### 3.2. Structural Designs and Modeling of the Bio-Engineered SC-TP-Gαq and SC-TP-Gαs

3D structures of human TP, Gαq, and Gαs are available; however, the Gαq and Gαs both lack complete terminal structures [17]. With the hope of understand how the expressed SC-TP-Gαq and SC-TP-Gαs perform dual functions (binding to a ligand and G-protein-mediating signaling) on a single polypeptide chain, it is essential to generate comprehensive 3D structural models. Thus, we started from construction of the terminal structure of TP and Gαq/s. The C-terminal 10 residues of TP, as well as the N-terminal 12 residues of Gαq or Gαs which both lack 3D structural coordinates, were created by homology modeling (Figure 1). After energy minimization, the C-terminus of TP was linked to the N-terminus of Gαq or Gαs, respectively. Energy minimization for the entire linked single-chain structure and the final structures of SC-TP-Gαq and SC-TP-Gαs were obtained. The detailed steps are summarized in Figure 1.

### 3.3. Structural and Functional Relationship of the Two Domains of TP and Gα within a Single Polypeptide Chain of SC-TP-Gαq and SC-TP-Gαs in Respect to the Cell Membrane

The primary structures of the designed single-chain SC-TP-Gαq and SC-TP-Gαs each contain sites with two functions: ligand binding and G-protein binding within a single polypeptide chain (Figure 2A). MK cells must be correctly formed to have dual function, including proper mRNA translation, single-chain protein expression, and post-translational modification. After membrane incorporation, SC-TP-Gα binds to TXA_2_ on extracellular/membrane domains and Gβγ on intracellular domains (Figure 2B). The membrane separating the two binding sites to avoid steric hindrance between the ligand and Gβγ binding sites on MKs and platelet cell membranes is illustrated in Figure 2B. The distance exceeding 30 Å between the ligand and Gβγ binding sites helps prevent steric hindrance, provided that the SC-TP-Gα protein undergoes proper folding and membrane integration (Figure 2C). In this setting, using TXA_2_ to mediate Ca^++^ signaling (via SC-TP-Gαq) and cAMP signaling (via SC-TP-Gαs) becomes feasible.

### 3.4. Design of Nucleus-Free PLPs to Deliver Recombinant SC-TP-Gαq or SC-TP-Gαs to Regulate Platelet Functions

For non-self-derived cell-based recombinant protein and gene therapy, one of the primary concerns is the potential contamination of genetic materials from the donor cell nucleus. To tackle this issue, we used the nucleus-free cellular delivery system of platelet-linked particles (PLPs). The procedures to produce the clinically applicable PLPs from human pluripotent cells and MK cells have been previously reported [17]. However, the utilization of PLPs as a carrier for recombinant GPCR and G-protein complexes has not been established. The procedures related to the transfection of cDNA plasmids of SC-TP-Gαq and SC-TP-Gαs to the immature MKs and to the generation of PLPs expressing the recombinant SC-TP-Gαq and SC-TP-Gαs are outlined in the Figure 3.

Our expectation is that bio-engineered SC-TP-Gαq and SC-TP-Gαs, which possess dual functions, could initially be expressed on MK cells and subsequently transferred to the PLPs. In this case, PLPs with membrane-bound SC-TP-Gαq should bind to the endogenous TXA_2_ and Gβγ within the same molecule and further promote Gαq-Ca^++^ signaling as an anti-bleeding agent (Figure 2, left). In contrast, PLPs with SC-TP-Gαs on the surface should bind to the same TXA_2_ and Gαs but redirect the PLPs to Gαs-cAMP signaling as an anti-thrombotic agent (Figure 2D, right). This design represents the first attempt to explore a method using identical endogenous ligands to regulate two opposite signaling mechanisms on platelets.

### 3.5. Construction of cDNA Vectors and Transfection of MK Cells Using Electroporation

The process of cDNA encoding the human SC-TP-Gαq or SC-TP-Gαs (Figure 4A) was established, and then each was further sub-cloned into human cell expression vector, pcDNA3.1, using the Ecor1 and Xho1 sites incorporated into the vector. The constructed vector is suitable for the transfection of human MKs in suspension. Overexpression of SC-TP-Gα on MK cells is the initial step to generate platelets carrying the expressed SC-TP-Gαq and SC-TP-Gαs to control platelet function. Here, we chose an MK cell line, Meg-01, as a model to generate our designed PLPs [16] (Figure 4B). The Meg-01 cell line was derived from a patient with chronic myelogenous leukemia, which has similar properties to human MKs. The well-characterized Meg-01 cell line obtained from ACCT was cultured in a suspension medium with the pcDNA3.1 vector (Figure 4B). Transfections of the Meg-01 cells using each cDNA vector were performed by a sandwich electroporation system with modification suitable for Meg-01 cells (Figure 4C). The best ratios of cDNA concentrations and electric pulse doses were established by a titration study.

### 3.6. Transient Gene Expression of SC-TP-Gαq and SC-TP-Gαs on MK Cell Lines

We used three steps to generate PLPs carrying SC-TP-Gαq and SC-TP-Gαs from immature MKs, which are: (A) transient expression; (B) generation of a stable expression cell line; and (C) induction of immature MKs to release PLPs. The transient gene expression, which is the first step toward successful production of the engineered PLPs, was characterized by microscopy and Western blot analysis. The effects of the heterogeneity of MK cells were not significantly changed by comparison before and after the cDNA transfection (Figure 5A). When comparing Meg-01 overexpression of SC-TP-Gαq or SC-TP-Gαs in suspension, both the immature and unaggregated characteristics remained similar, including the different signaling molecules introduced into the cells (Figure 5B). Transient expression of SC-TP-Gαq and SC-TP-Gαs on the immature Meg-01 cells was confirmed by Western blot analysis (Figure 5C). These results indicate that expression of SC-TP-Gαq and SC-TP-Gαs in the Meg-01 cells has been successfully achieved. The transfected immature Meg-01 cells are suitable for establishing cell lines with stable expression of SC-TP-Gαq or SC-TP-Gαs.

### 3.7. Screening of the Immature Meg-01 Cells Stably Expressing SC-TP-Gαq to Generate PLPs Carrying SC-TP-Gαq

The immature Meg-01 cells, which were transfected with the cDNA of SC-TP-Gαq as depicted in Figure 5, were subsequently employed for screening of the cell line stably expressing SC-TP-Gαq. First, the transfected immature Meg-01 cells with transient expression of SC-TP-Gαq (Figure 5) were treated with the culture medium containing G-418 as a selecting reagent (Figure 6A). Following three weeks, the untransfected Meg-01 cells were eliminated by the G-418, and the cell line stably expressing SC-Gαq (MK-SC-TP-Gαq cell line) remained in the suspension. A typical stable cell is shown in Figure 6A. In the subsequent step, the immature MK-SC-TP-Gαq cell line was induced to maturation by the addition of megakaryocyte growth and development factors (MGDFs) and Thrombopoietin (TPO) [16] (Figure 6B). The images collected from the steps to generate PLPs carrying SC-TP-Gαq are summarized in Figure 6A,B. The typical mature Meg-01 cells, which release the PLPs carrying the expressed SC-TP-Gαq (PLP-SC-TP-Gαq), are shown in Figure 6B. The immature stable Meg-01 line and mature PLPs expressing the recombinant SC-TP-Gαq were confirmed by Western blot analysis (Figure 6C).

### 3.8. Screening of Meg-01 Cells Stably Expressing SC-TP-Gαs to Generate PLPs Carrying SC-TP-Gαs

The immature Meg-01 cells transfected with the cDNA of SC-TP-Gαs, as shown in Figure 5, were used for screening of the cell line stably expressing SC-TP-Gαs. First, the transfected Meg-01 cells expressing SC-TP-Gαs (Figure 5) were treated with the culture medium containing G-418 for three weeks. The untransfected Meg-01 cells were eliminated by the G-418, and the cell line stably expressing the SC-TP-Gαs (MK-SC-TP-Gαs) was obtained. The immature MK-SC-TP-Gαs cell line was subsequently induced to maturation with the ability of release of PLPs by the addition of MGDFs and TPO. The typical mature Meg-01, which releases PLPs carrying the expressed SC-TP-Gαs (PLP-SC-TP-Gαq), is shown in Figure 7a. PLP-SC-TP-Gαs expressing the recombinant SC-TP-Gαs was confirmed by Western blot analysis (Figure 7b).

### 3.9. Establishing Highly Sensitive Flow Cytometry Using Double-Stained Human Platelets for Identification of the Biological Activities of PLP-SC-TP-Gαq and PLP-SC-TP-Gαs

Flow cytometry was performed to determine the biological activities of PLP-SC-TP-Gαq and PLP-SC-TP-Gαs by monitoring their effects on human platelet aggregation. In this assay, two antibodies—the anti-platelet markers CD41a and CD42b—were labeled with FITC and APC to generate the FITC-CD41a antibody and the APC-CD42b antibody, respectively. Human platelet-rich plasma (PRP) was prepared from human platelets obtained from a local blood bank. Half of the PRP sample was stained with the FITC-CD41a antibody and the other half with the APC-CD42b antibody. The samples were mixed, and 3 µM of arachidonic acid was added to induce aggregation. The sample was analyzed by flow cytometry. The signals, double stained by both CD41a and CD42b, were used to determine platelet aggregation (Figure 8). The positive and negative results for each staining were analyzed using three groups of samples. In the first group, platelets without any staining treatments (FITC-, APC-) were used as the negative control. The area with cells present was defined as the Q3 Quadrant (Figure 8a). The second group was composed of a mixture of FITC-CD41a-stained and -negative platelets. After analysis by flow cytometry, two populations were present (Figure 6b. Left: FITC-; Right: FITC+). The area containing FITC-positive cells was defined as the Q4 Quadrant (Figure 8b). The third group was composed of APC-conjugated CD42b-stained platelets mixed with CD42b-negative platelets. Following flow cytometry analysis, two populations were present: the upper population of APC-positive platelets in the area defined as the Q1 Quadrant (Figure 8c) and the Q2 Quadrant which described the double-staining population, indicating the aggregation of platelets.

### 3.10. Determination of the Effects of the Nucleus-Free PLP-SC-TP-Gαq and PLP-SC-TP-Gαs on Regulating Platelet Aggregation Using the Double-Stained Flow Cytometry

Based on the signaling cascades mediated by SC-TP-Gαq and SC-TP-Gαs, we hypothesized that the PLPs expressing SC-TP-Gαq could further promote platelet aggregation, while the PLPs expressing SC-TP-Gαs could inhibit platelet aggregation. To verify our hypothesis, three groups of platelet mixtures were analyzed by flow cytometry to determine the effects of SC-TP-Gαq and SC-TP-Gαs on platelet aggregation activities. The first group contained a mixture of 200 µL human PRP and 200 µL nucleus-free PLPs released by WT Meg-01 cells. After inducing aggregation by AA, roughly 67% percent aggregation was observed in the Q2 double-staining Quadrant (Figure 9a). The second group was composed of 200 µL PRP mixed with 200 µL nucleus-free PLP-SC-TP-Gαq. The aggregative activity was induced by the addition of AA, and an increased aggregation percentage (~74%) was identified (Figure 9b). This is explained by the ability of PLP SC-TP-Gαq to mediate calcium signaling, which could further increase the percentage of platelet aggregation. The last group was composed of a mixture of 200 µL PRP with 200 µL nucleus-free PLP-SC-TP-Gαs. The aggregative reaction was also triggered by the addition of AA, which dramatically decreased the aggregation percentage to 9% (Figure 9c). This resulted from cAMP signaling mediated by SC-TP-Gαs which inhibited platelet aggregation. Three trials were conducted in each group. Statistical analysis of the results led to the conclusion that the PLP-SC-TP-Gαq protein complex has biological activity to promote platelet aggregation, which might be beneficial in anti-bleeding. In contrast, PLP-SC-TP-Gαs has activity to dramatically prevent platelet aggregation, which may be beneficial in treating thrombotic diseases (Figure 9d).

### 3.11. Working Model of PLP-SC-TP-Gαq and PLP-SC-TP-Gαs Regulation of Human Platelet Aggregation

A working model diagram is described in Figure 10. In the presence of PLP-SC-TP-Gαq, TXA_2_ produced from AA as a substrate catalyzed by COX-1 coupled to TXAs in human platelets binds to TP (Figure 10A) and SC-TP-Gαq on PLPs (Figure 10B). This triggers Ca^++^ signaling on both cells, leading to overlapping platelet aggregation (Figure 10B). In contrast, only part of TXA_2_ binds to TP on human platelets, triggering platelet aggregation. Another part of TXA_2_ binds to SC-TP-Gαs on PLPs triggering cAMP-mediated anti-platelet aggregation, thereby effectively inhibiting human platelet aggregation (Figure 10C). To explain further, SC-TP-Gαs expressed on PLPs inhibited platelet aggregation through dual effects: one decreased the amount of TXA_2_ binding to human platelets, and the other triggered cAMP signaling to prevent human platelet aggregation.

## 4. Discussion

Our study is consistent with previous reports indicating that PLPs can be generated from platelet precursor cells and stem cells [25,26,27]. The therapeutic impact of PLPs has been well documented; for example, PLPs have shown the ability to improve wound healing [28,29]. Several groups have suggested that non-donor-based platelets derived from in vitro-grown megakaryocytes may serve as supplements to donor-derived platelets [30]. While the therapeutic potentials of nucleus-free PLPs have been previously demonstrated, the majority of the studies are limited to the applications of “mimicking platelet functions.”

Here, we broaden the application by testing PLPs as a recombinant protein drug delivery system. Specifically, we focused on establishing the unique therapeutic potentials as a nucleus-free cell recombinant membrane protein therapeutic agent and a novel cell-based GPCR-G-protein signaling therapy. In this study, we discovered advantages to the PLP-based recombinant membrane protein delivery system. First, PLPs do not have nuclei. This prevents potential contamination of nucleic acid and genetic DNAs. PLPs contain fewer proteins and enzymes compared with other therapeutic cells, which limits unwanted side effects. Because of the small size of PLPs, they are suitable for many applications, including IM, IP, and IV infusion. In contrast to other cell-based delivery systems, such as intramuscular injection of stem cells, the IV infusion of PLPs does not cause possible acute inflammatory reactions. In other cell-based delivery systems, the elimination of aggregative cells creates an issue for IV infusion. However, PLPs (suspension cells) are fully suitable for IV infusion. Sources for these clinical PLPs are no longer limited. Recently, a research group established a megakaryocyte cell line which can produce clinically applicable PLPs [16,17].

As a class of recombinant protein drug delivery carriers, PLPs are needed to express the desired recombinant proteins within the cells. However, PLPs themselves do not have an existing system to make recombinant proteins. In this study regarding the transfection of PLP precursor cells, MKs were designed to solve this issue. Immature MKs were transfected using cDNA plasmids to express the desired recombinant proteins. After maturation of the MKs, the expressed recombinant proteins within the cells were passed on to the released PLPs. We successfully utilized the Meg-01-PLP system as a model in the study. After the establishment of stable immature Meg-01 cell lines expressing the desired recombinant proteins, the MKs were induced to maturation to release PLPs with the expressed recombinant protein. Therefore, we demonstrated that the nucleus-free PLPs carrying the recombinant protein have potential to become a novel tool to deliver recombinant proteins in therapeutics.

Currently, membrane proteins have limited potential in therapeutics due to their insolubility and aggregation potential. However, membrane proteins, such as GPCRs, are directly involved in conducting cellular activities. We hypothesized that the PLP-recombinant protein approach is suitable to carry many recombinant proteins, including membrane-bound proteins. Here, we used SC-TP-Gαq and SC-TP-Gαs with seven trans-membrane domain proteins as a model to test the PLP system’s ability to transport membrane proteins. The study confirmed that the PLP-recombinant membrane protein system has potential to regulate cellular functions. For example, PLPs expressing SC-TP-Gαs could serve as a “cleaner” with anti-thrombotic effects to continually absorb the excessive TXA_2_ in circulation. PLPs carrying SC-TP-Gαq could be used to promote platelet aggregation as an anti-bleeding agent in diverse bleeding situations. By applying PLP-SC-TP-Gαq to the bleeding site, the TXA_2_, released by the bleeding tissue, could trigger downstream Gαq-calcium signaling. This mimics the natural signaling cascades mediated by TP, promoting platelet aggregation and vasoconstriction. Conversely, under the same TXA_2_ stimulation, PLP-SC-TP-Gαs is able to specifically convert downstream signaling into Gαs-cAMP signaling, which opposes natural signaling activities mediated by the TP receptor. This inhibits platelet aggregation, mediates vasodilation, promotes anti-thrombotic effects, and reduces the risk of strokes and various thrombotic diseases.

In conclusion, these findings show that nucleus-free PLPs derived from the transfected MKs can be used as a delivery vehicle for recombinant membrane proteins by controlling cell signaling. Thus, the PLP-recombinant protein system has potential for development into a novel and advanced therapeutic tool to carry a variety of insoluble membrane proteins. For example, it could be used as a therapeutic agent to counter vascular thrombosis by introducing PLP-SC-TP-Gαs into the vascular system. Furthermore, this research suggests that the PLP system can be manipulated as a regulatory and therapeutic tool to control and even reverse the existing functions of GPCRs, which may become a future method to regulate the functions of cellular GPCRs, not including traditional drugs as agonists and antagonists of the GPCRs.

## Figures and Tables

**Figure 1 cells-12-02775-f001:**
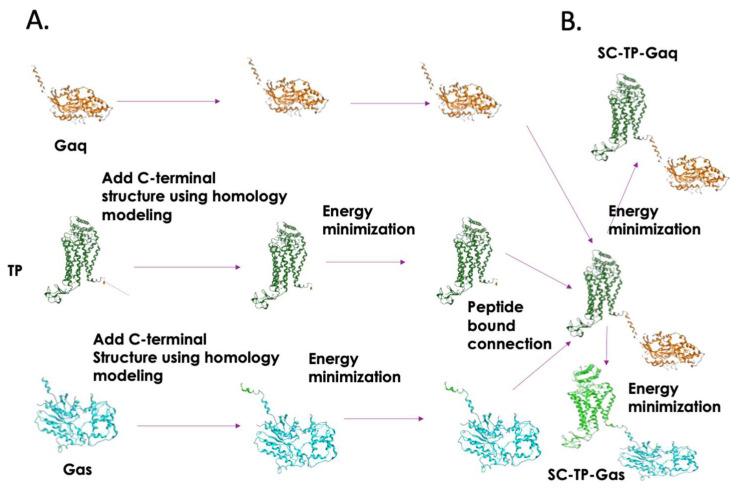
Procedure for the structural design of SC-TP-Gα expression in MK-PLPs. (**A**) Modeling of unavailable 3D structures of the C-terminal region of TP (PDB ID: 6IIU) and N-terminal region of Gαq (PDB ID: 4GNK) and Gαs (PDB ID: 6R4O) by adding the terminal structures to the TP and Gαq/s and (**B**) the covalently linked TP C-terminus to the Gαq or Gαs N-terminus to form the SC-TP-Gαq or the SC-TP-Gαs. Energy minimization was applied to each step to obtain final structures. The 3D structures used for the models are crystal structures of human TP (PDB ID: 6IIU) [23], Gαq (PDB ID: 4GNK) [24], and Gαs (PDB ID: 6R4O) [21]. The PLPs carrying SC-TP-Gαq (**left**) and SC-TP-Gαs (**right**) were schematically presented.

**Figure 2 cells-12-02775-f002:**
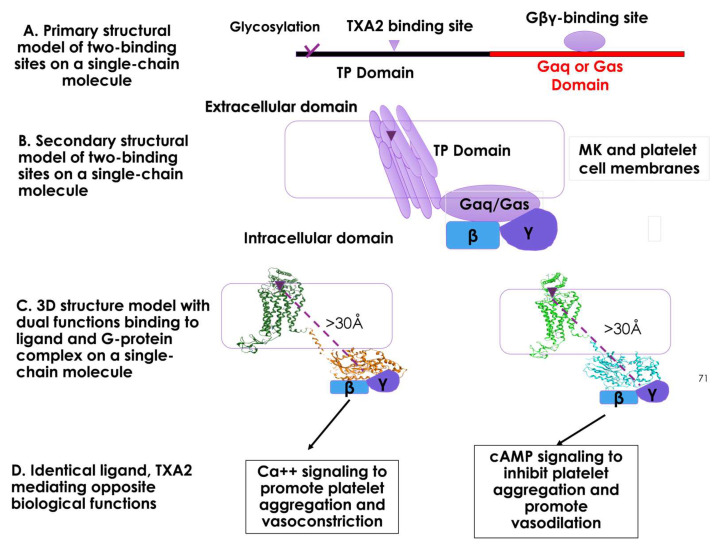
An illustration of the designs and models of the SC-TP-Gα with dual functions within a single peptide chain. (**A**) Model illustrating the primary structure of the SC-TP-Gα. (**B**) Model of the secondary structure of the SC-TP-Gα. (**C**) Distance of the two binding sites on the SC-TP-Gα. (**D**) Schematic presentation of the production of nucleus-free PLPs expressing ST-TP-Gαq or SC-TP-Gαs with opposite biological functions to regulate platelet functions.

**Figure 3 cells-12-02775-f003:**
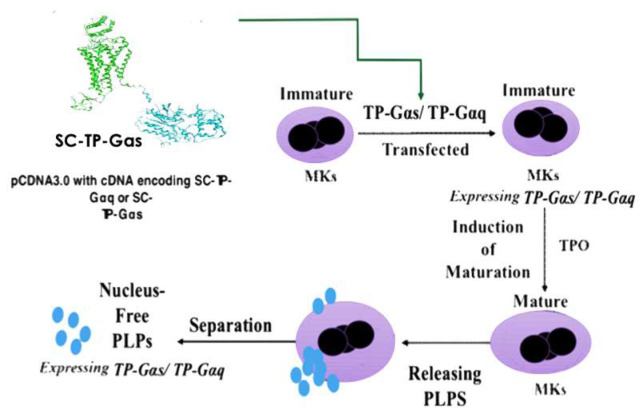
Design of nucleus-free PLPs generated from MK cells to deliver recombinant SC-TP-Gα proteins with therapeutic potentials. MK cells were present in purple color, and the nucleus-free PLPs were represented by small blue dots.

**Figure 4 cells-12-02775-f004:**
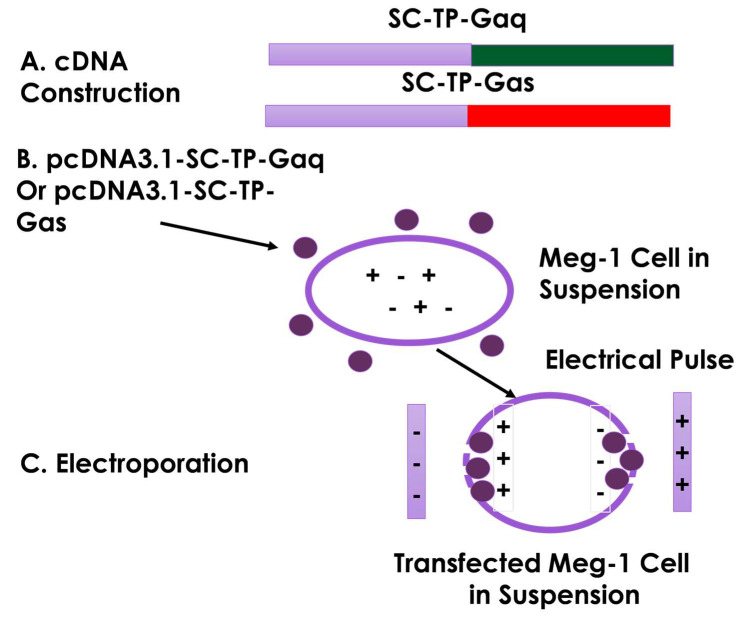
Procedure for transfection of the Meg-01 cells in suspension using electroporation. (**A**) cDNAs of SC-TP-Gαq or SC-TP-Gαs were constructed using a PCR approach. (**B**) pcDNA3.1 vector with cDNA of SC-TP-Gαq or SC-TP-Gαs was prepared. (**C**) pcDNA3.1 vector was transfected into the Meg-01 cell in suspension by electric pulse in a 2 mm cuvette (set voltage: 600 V, pulse length: 100 μs–200 μs, number of pulses: 2). Meg-01 cells were represented in purple circle. The purple lines represented the electroporation cuvette wall.

**Figure 5 cells-12-02775-f005:**
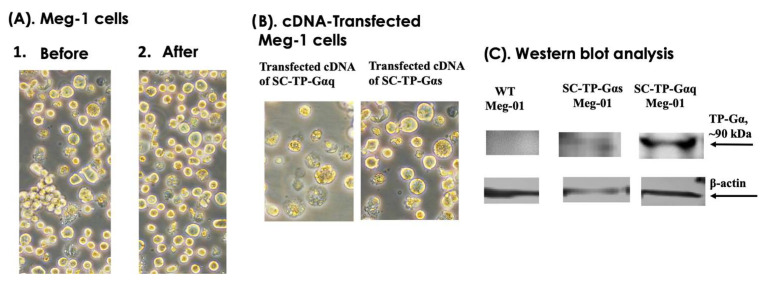
(**A**) Suspension culture of Meg-01 cells before and after cDNA-transfection. (**B**) Suspension culture of Meg-01 cells transfected with SC-TP-Gαq (**left**) and SC-TP-Gαs (**right**). (**C**) Western blot analysis. The expressed SC-TP-Gαs (lane 2) and SC-TP-Gαq (lane 3) on Meg-01 cells are shown.

**Figure 6 cells-12-02775-f006:**
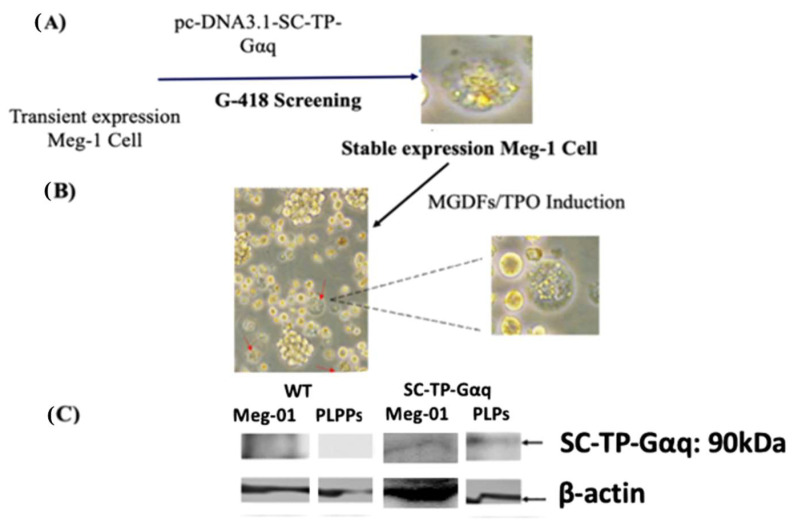
Production of PLPs expressing SC-TP-Gαq from immature Meg-01 cells. (**A**) Stable cell line screening; (**B**) PLP production; and (**C**) Western blot analysis using 10% SDS-PAGE and anti-human TP antibody. Red arrows are represented the typical mature Meg-01 cells, which release the PLPs carrying the expressed SC-TP-Gαq.

**Figure 7 cells-12-02775-f007:**
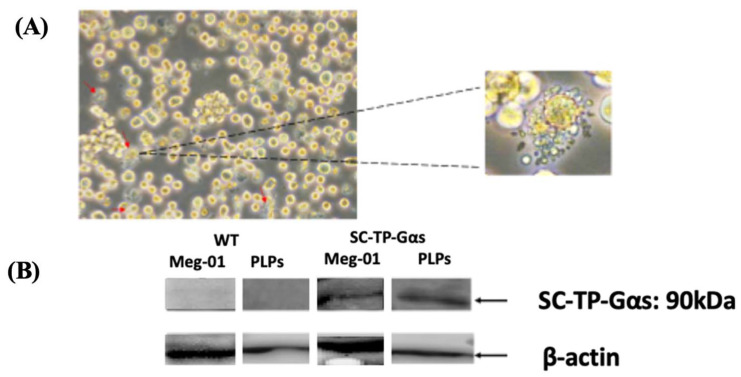
Induction of maturation of the Meg-01 cells expressing SC-TP-Gαs. (**A**) The pro-platelet structure was also identified under microscope. (**B**) The expression of SC-TP-Gαq in PLPs was also confirmed by Western blot.

**Figure 8 cells-12-02775-f008:**
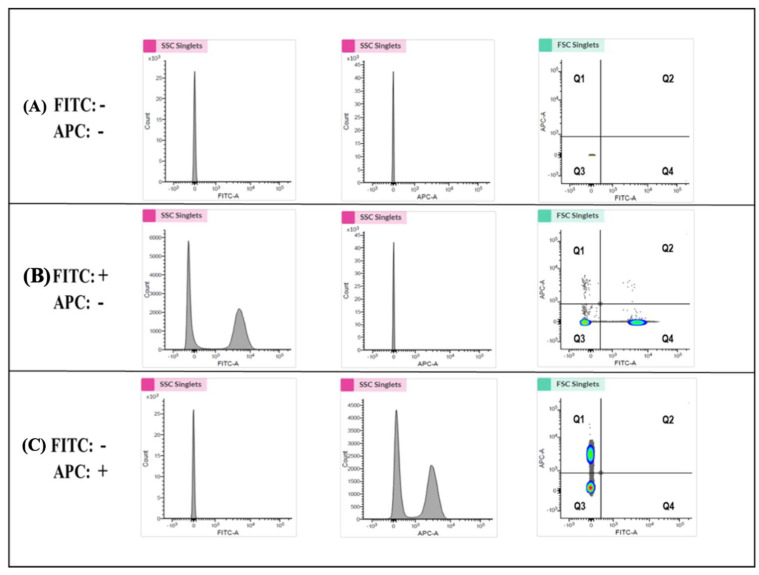
Determination of the specific quadrant to indicate platelet aggregation activities. (**A**) The analysis of platelets without staining was shown. The Q3 Quadrant represents the negative platelets without fluorescence. (**B**) The platelets stained with the FITC marker were mixed with the FITC-negative platelets. The shown left population represents the FITC-negative platelets, and the right population (Q4 Quadrant) represents the FITC-positive but APC-negative platelets. (**C**) The mixture of APC marker-stained platelets and APC-negative platelets was analyzed. The bottom population represents the ACP-negative platelets, and the upper one (Q3) represents the APC-positive but FITC-negative platelets. The Q4 Quadrant, which represents the double staining by both FITC and APC, indicates platelet aggregation activity.

**Figure 9 cells-12-02775-f009:**
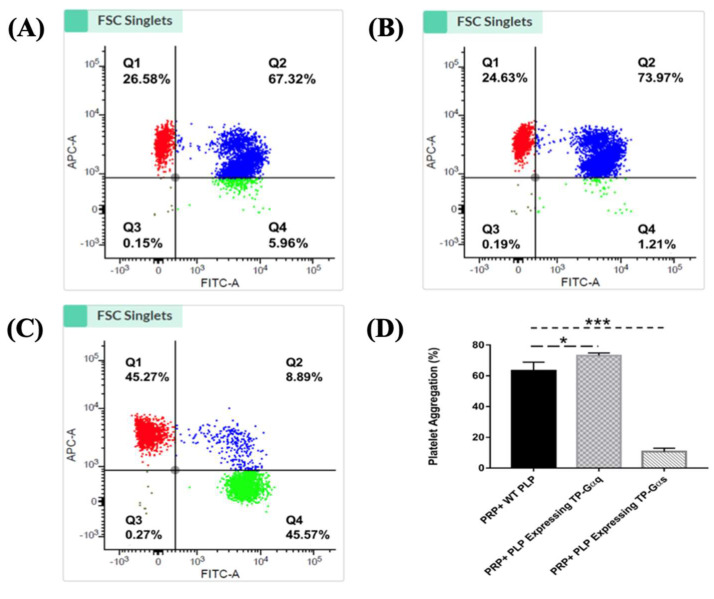
Identification of the PLPs expressing the protein complexes in platelet aggregation assays. (**A**) A quantity of 200 µL PRP was mixed well with 200 µL WT PLPs. The mixed sample was divided equally into two groups: one was stained with the FITC-conjugated CD41a antibody, and the other was stained with the APC-conjugated CD42b antibody. After staining, the two groups were mixed again thoroughly, and AA was added to induce aggregation (3 µM). Then, the sample was analyzed by flow cytometry. About 67% of aggregation was recognized. (**B**) A quantity of 200 µL PRP was mixed completely with 200 µL PLPs expressing SC-TP-Gαq. Then, the above procedures were repeated. The flow cytometry analysis indicated about 74% of aggregation. (**C**) The mixture of 200 µL PRP and 200 µL PLPs expressing SC-TP-Gαs was used to repeat the above steps. Around 9% of aggregation was demonstrated in this group. (D) Statistical analysis of platelet aggregation. * *p* < 0.05 *** *p* < 0.001.

**Figure 10 cells-12-02775-f010:**
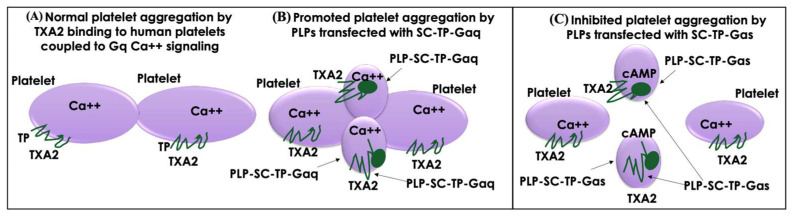
Models of the effects of the PLP-SC-TP-Gαq and PLP-SC-TP-Gαs on platelet aggregation. (**A**) In the absence of PLPs, TXA_2_ binds to TP on the platelet surface, which couples to Gq and mediates Ca^++^ signaling and platelet aggregation. (**B**) In the presence of PLPs expressing SC-TP-Gαq, TXA_2_ binds to TP on platelets and SC-TP-Gαq on PLPs, which enhances Ca^++^ signaling and overlaps platelet and PLP aggregation. (**C**) In the presence of PLPs expressing SC-TP-Gαs, TXA_2_ binds to TP on platelets and SC-TP-Gαs on PLPs, which increases cAMP signaling to inhibit platelet and PLP aggregation.

## Data Availability

All of the data published are available for public.

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
