# Peer review of "Reprogramming Megakaryocytes for Controlled Release of Platelet-like Particles Carrying a Single-Chain Thromboxane A2 Receptor-G-Protein Complex with Therapeutic Potential"

_cells, 2023, doi:10.3390/cells12242775_

Round 1

Reviewer 1 Report

Comments and Suggestions for Authors

Generally, the topic is interesting and the paper was well-designed. However, there are a few sections of the manuscript that need to be improved as follows:

1. The Introduction: the background information is not adequate enough to provide the readers with basic knowledge about the topic, it needs to richer than it is

2. Result section: the image quality of all the figures is very for a reputable journal like Cell. Thus, the quality needs to be improved.

3. Discussion: there are not enough citations despite quoting many references in the discussion section. The section needs to be supported with more evidence and of course appropriate cited references.

Comments on the Quality of English Language

The paper needs to be edited by a native speaker or an expert, for there are few errors to be addressed

Author Response

  1. The Introduction: the background information is not adequate enough to provide the readers with basic knowledge about the topic, it needs to richer than it is.

We have improved the introduction with adding more detail of the background.

  1. Result section: the image quality of all the figures is very for a reputable journal like Cell. Thus, the quality needs to be improved.

Based on your advice, we have made changes to the images, increasing their clarity and resolution.

  1. Discussion: there are not enough citations despite quoting many references in the discussion section. The section needs to be supported with more evidence and of course appropriate cited references.

With the intention of accomplishing more detail of the background, we have cited more articles to support our manuscript.

Reviewer 2 Report

Comments and Suggestions for Authors

Title: Reprogramming megakaryocyte gene expression to release platelet-like particles with opposite biological activities: carrying single-chain thromboxane A2 receptor-G-protein complex with therapeutic potentials

In the research article, the authors demonstrate single-chain fusion proteins linked TP receptor to Gαq or Gαs G-protein subunit and their stable expression in megakaryocytes and role in platelet aggregation. 

 The comments are below:

1.  Authors did not mention TP isoforms in the manuscript.  

2.  The authors should perform Flow cytometry analysis to establish the location of TP- Gαq or Gαs in a stable overexpression system.

3.  Figures are not clear.  

4.   Authors performed homology modeling for TP C-terminal and Gq or Gs N-terminal. What is the possibility of TP C-ter and G proteins C- ter interaction and downstream signaling?

5.  Did the authors try to see the functional role of this complex with any TP agonist like  U46619?

6.  How can this complex be affected by homologous or heterologous glycosylation?   Did the authors try to use GntI system and see its effect?

7. What is the stability of PLPs after their secretion from mature MK cells?

8.  Did the authors use the whole TP and Gq proteins to bioengineer SC-TP-Gq or, similarly, for SC-TP-Gs?  What is the possibility of conformational changes with this ~90KDa complex, and how can this change the pharmacokinetics with any agonist?

9. The authors should add the catalog number and manufacturer’s names for all antibodies used in the method and materials section.

10. Line 172: Correct Gby to Gβγ unit.

11.Line 178: What is GSC in TP-GSC-TP-Gαs

12. Figure 4: Add electroporation conditions in legend e.g. Volt and time and how many times.

13. Fig 6C: Beta-actin western blot is not clear. It looks like a smear. The authors can normalize the expression with Beta-actin by densitometry and show it with a Bar diagram.

14. Overall the western blots are of poor quality.

An interesting approach was used for the research, but manuscript requires major revision.

Comments on the Quality of English Language

Overall the language and sentences are not well constructed.

Author Response

  1. Authors did not mention TP isoforms in the manuscript.  

TP have two isoforms, Alpha, and Beta. In this study, we focus on the Alpha which is the major isoform in human blood. We have added this information in the revision (see 2.3. Designing SC-TP-Gαq and SC-TP-Gαs).

  1. The authors should perform Flow cytometry analysis to establish the location of TP- Gαq or Gαs in a stable overexpression system.

The expressed TP- Gαq and TP-Gαs localized on cell membrane have been identified (see refe.24).

  1. Figures are not clear.  

Thank you for your advice, we have increased the quality of all the figures.

  1. Authors performed homology modeling for TP C-terminal and Gq or Gs N-terminal. What is the possibility of TP C-ter and G proteins C- ter interaction and downstream signaling?

Yes, we are conducting the homology modeling study using MOE software. However, due to the spatial impediment between TP and G alpha q/s, the only viable method to establish our enzyme link is by connecting TP C-terminal and Gq or Gs N-terminal. Other forms of TP-G protein hybrids could not be successfully cloned.

  1. Did the authors try to see the functional role of this complex with any TP agonist like U46619?

Very good suggestion, the functions of TP- Gα has been identified by a similar TP agonist IBOP previously (see ref. 24).

  1. How can this complex be affected by homologous or heterologous glycosylation?   Did the authors try to use GntI system and see its effect?

The major glycocilation site for TP on the extracellular N-terminal region. We keep intact for the N-terminal regions of TP-Gaq and TP-Gas. Thus, the they shall have similar glycosylation as wild type.

  1. What is the stability of PLPs after their secretion from mature MK cells?

Human platelets typically have a lifespan in the bloodstream, approximately 7 to 10 days.  

Our designed PLPs expressed by the human MK cells should have the same characteristic.

  1. Did the authors use the whole TP and Gq proteins to bioengineer SC-TP-Gq or, similarly, for SC-TP-Gs?  What is the possibility of conformational changes with this ~90KDa complex, and how can this change the pharmacokinetics with any agonist?

The structures of SC-TP-Gαs/q were generated using MOE 2022 software. We filtered through various conformational possibilities provided by the software and selected the one with the lowest energy as our homomodeling model. The conformation changes have been mimicked by the software. As the design strategy of our enzymelink is linked the two domains, so the protein structure of TP and Gs/q will not influence.

  1. The authors should add the catalog number and manufacturer’s names for all antibodies used in the method and materials section.

Thank you for your advice, we have added all the catalog number and manufacturer’s names in the manuscript.

  1. Line 172: Correct Gby to Gβγ unit.

Yes, we have corrected the spelling problems.

11.Line 178: What is GSC in TP-GSC-TP-Gαs

It is a typo. We have corrected via TP-GSC-TP-Gα to SC-TP-Gα.

  1. Figure 4: Add electroporation conditions in legend e.g. Volt and time and how many times.

We have included the specific parameters for electroporation in the manuscript.

  1. Fig 6C: Beta-actin western blot is not clear. It looks like a smear. The authors can normalize the expression with Beta-actin by densitometry and show it with a Bar diagram.

Thank you for the suggestion, we have improved the quality using densitometry.

  1. Overall the western blots are of poor quality.

Thank you for your suggestion, we will improve the quality.